# Sexually Dimorphic Alterations in the Transcriptome and Behavior with Loss of Histone Demethylase *KDM5C*

**DOI:** 10.3390/cells12040637

**Published:** 2023-02-16

**Authors:** Katherine M. Bonefas, Christina N. Vallianatos, Brynne Raines, Natalie C. Tronson, Shigeki Iwase

**Affiliations:** 1Department of Human Genetics, Michigan Medicine, University of Michigan, Ann Arbor, MI 48109, USA; 2Neuroscience Graduate Program, University of Michigan, Ann Arbor, MI 48109, USA; 3Genetics and Genomics Graduate Program, University of Michigan, Ann Arbor, MI 48109, USA; 4Department of Psychology, College of LS&A, University of Michigan, Ann Arbor, MI 48109, USA

**Keywords:** chromatin regulators, x-linked intellectual disability, learning and memory, neurodevelopmental disorders, histone demethylase

## Abstract

Chromatin dysregulation has emerged as a major hallmark of neurodevelopmental disorders such as intellectual disability (ID) and autism spectrum disorders (ASD). The prevalence of ID and ASD is higher in males compared to females, with unknown mechanisms. Intellectual developmental disorder, X-linked syndromic, Claes-Jensen type (MRXSCJ), is caused by loss-of-function mutations of lysine demethylase 5C (*KDM5C*), a histone H3K4 demethylase gene. KDM5C escapes X-inactivation, thereby presenting at a higher level in females. Initially, MRXSCJ was exclusively reported in males, while it is increasingly evident that females with heterozygous *KDM5C* mutations can show cognitive deficits. The mouse model of MRXSCJ, male *Kdm5c*-hemizygous knockout animals, recapitulates key features of human male patients. However, the behavioral and molecular traits of *Kdm5c*-heterozygous female mice remain incompletely characterized. Here, we report that gene expression and behavioral abnormalities are readily detectable in *Kdm5c*-heterozygous female mice, demonstrating the requirement for a higher KDM5C dose in females. Furthermore, we found both shared and sex-specific consequences of a reduced KDM5C dose in social behavior, gene expression, and genetic interaction with the counteracting enzyme KMT2A. These observations provide an essential insight into the sex-biased manifestation of neurodevelopmental disorders and sex chromosome evolution.

## 1. Introduction

Mutations in methyl histone regulators are overrepresented in neurodevelopmental disorders (NDDs) such as intellectual disabilities and autism spectrum disorders [1,2,3,4,5]. The prevalence of these conditions is higher in males than in females: approximately 1.5-fold for intellectual disability and 4-fold higher for autism spectrum disorders [6,7]. Biological differences such as sex chromosomes and hormones may contribute to sex-biased prevalence. However, the diagnosis of these conditions is based on the assumption that behavioral manifestations of a specific brain malfunction are the same between the human sexes. The validity of the assumption is questionable given the above biological differences, and the incorrect assumption could contribute to the sex-biased prevalence [8]. Thus, understanding the sex-specific impact of methyl histone dysregulation is necessary to grasp the prevalence and manifestation of NDDs accurately. 

One way to measure the impact of sex on the pathophysiology of NDDs is to compare females and males in animal models with a defined genetic lesion. Mouse models of NDD associated with histone methylation dysregulation have demonstrated a critical role of methyl writer [9,10,11,12] and eraser enzymes [13,14,15] in normal cognitive development. Some studies have examined both sexes in mouse models [11,16,17,18,19]. When ATRX (Alpha-thalassemia/mental retardation-X gene), an X-linked chromatin remodeling enzyme that recognizes Histone 3 Lysine 9 methylation, was deleted, the mice exhibited sexually dimorphic behavior and gene expression changes [17,18]. However, most studies either solely used males or failed to disaggregate data by sex. As a result, little is known about the etiology, pathogenesis, and manifestations of NDDs in females. 

Lysine demethylase 5C (*KDM5C*) represents an excellent case for exploring the sex-specific impact of methyl histone dysregulation. *KDM5C* is X-linked and encodes an enzyme, which removes methylation specifically from histone H3 lysine 4 (H3K4me) [20,21]. Loss-of-function mutations in *KDM5C* cause intellectual developmental disorder, X-linked syndromic, Claes-Jensen type (MRXSCJ), characterized by shorter statures, cognitive deficits, autistic features, and aggressive behavior [22]. Most early *KDM5C*-MRXSCJ cases reported were males in whom the mutations were transmitted from one of the X chromosomes of carrier mothers. It is now clear, however, that some females with heterozygous *KDM5C* mutations also show cognitive deficits, albeit milder ones than males [23,24,25,26,27,28]. Notably, some female carriers are asymptomatic; thus, the penetrance is incomplete. In addition, *KDM5C* escapes X inactivation in most mammals, including humans and mice [29]. As a result, females have a higher *KDM5C* dose than males, and males can be cognitively normal despite the lower *KDM5C* dose. Taken together, some, if not all, females appear to require a higher *KDM5C* dose than males for normal brain development in humans with unknown mechanisms. 

Studies of *Kdm5c*-knockout mice further validated the differential requirement of *KDM5C* between the sexes. The complete loss of *Kdm5c* in hemizygous male mice (*Kdm5c*-KO) resulted in slightly smaller body size and behavioral abnormalities akin to MRXSCJ patients [13,14]. These include memory impairments and social behavior abnormalities, such as increased aggression [13,14]. In contrast, homozygous knockout females die in utero around embryonic day five [30], indicating that *KDM5C* is essential for embryogenesis in females but not in males. *Kdm5c-HET*erozygous (HET) female mice also display memory deficits [31]; however, the characterization of cognitive-social behaviors in female *Kdm5c-HET* mice is limited. Furthermore, gene expression changes in *Kdm5c-HET* mice remain unknown. 

The reversible nature of histone modifications can be a promising therapeutic target. For example, while *KDM5C* is an H3K4me eraser, *KMT2A* is an H3K4me writer, and *KMT2A* haploinsufficiency is associated with the NDD Weidemann–Steiner Syndrome (WDSTS) characterized by intellectual disability, developmental delay, hairy elbows, and short stature in both males and females [32,33]. Consistently, *Kmt2a* in mouse excitatory neurons is essential for normal learning and memory, anxiety regulation, H3K4me3 distributions, and gene expression of both sexes [11,12]. With male mice, we previously showed that *Kmt2a-Kdm5c* double mutations corrected many of the behavioral, cellular, and molecular deficits of the single mutants, either *Kdm5c*-KO or *Kmt2a-HET* [13]. However, it remains unknown if there is any sex difference in the *Kdm5c-Kmt2a* genetic interaction. 

In the present work, we defined shared and sex-specific gene expression changes in *Kdm5c-HET* female mice and determined the genetic interaction of *Kmt2a* and *Kdm5c* in female mouse behavior. Unlike the prior work that used three- to six-month-old adult mice for gene expression analysis [13,14] in this study, we analyzed postnatal day six (P6) mice, in which critical neural maturation takes place. We found that the female *Kdm5c-HET* forebrain showed clear misregulation in hundreds of genes, despite an intact *Kdm5c* allele, supporting the notion females require a higher *Kdm5c* dose through X-inactivation escape. Furthermore, both unique and shared genes were misregulated between the male and female mutants. Importantly, the sex differences and similarities extend to the behavior and the writer–eraser interaction. 

## 2. Materials and Methods

### 2.1. Mice 

Generation of the *Kdm5c*-KO allele was previously described by Cre-mediated deletion of exons 11 and 12 [14]. For sequencing experiments, the *Kdm5c*-KO allele was maintained on a mixed background of C57BL/6J and 129S1/SvImJ. Samples for sequencing were collected from 3 litters. Sex was determined through genotyping primers for *Uba1* on the X and Y chromosomes with the following primers: 5′-TGGATGGTGTGGCCAATG-3′, 5′-CACCTGCACGTTGCCCTT-3′. Deletion of *Kdm5c* was determined through the primers 5′-ATGCCCATATTAAGAGTCCCTG-3′, 5′-TCTGCCTTGATGGGACTGTT-3′, and 5′-GGTTCTCAACACTCACATAGTG-3′. For behavioral experiments, F1 hybrids were generated by crossing female mice carrying the *Kdm5c*-KO allele on a congenic C57BL/6J background with mice heterozygous for loss of *Kmt2a* (*Kmt2a*-HET) on a congenic 129S1/SvImJ background, as previously described [13]. All mouse studies complied with the protocols (PRO00010350: Iwase and PRO00008807: Tronson) by the Institutional Animal Care & Use Committee (IACUC) of The University of Michigan. 

### 2.2. mRNA-Seq 

The forebrain (hippocampus and cortex) was micro-dissected from postnatal day six (P6) mice. We used four animals per genotype. Total RNA was purified and DNAse-treated by the Qiagen RNEasy Midi Kit (Qiagen #75144) after homogenizing the samples in Buffer RLT with an electric homogenizer. Purified RNA was sent to Novogene for sample quality control and cDNA library preparation by poly A enrichment. Libraries were prepared using the NEBNext^®^ Ultra™ II Directional RNA Library Prep Kit with oligo-dT priming and sequenced through the Illumina NovaSeq 6000 sequencing platform to generate paired-end 150 base-pair reads. After ensuring read quality via FastQC (v0.11.8), reads were then mapped to the mm10 *Mus musculus* genome (Gencode) using STAR (v2.5.3a), during which we removed duplicates and kept only uniquely mapped reads. Count files were generated by FeatureCounts (Subread v1.5.0), and BAM files were converted to bigwigs using deeptools (v3.1.3) and visualized by the UCSC genome browser. RStudio (v3.6.0) was then used to analyze counts files by DESeq2 (v1.26.0) [34] to identify differentially expressed genes (DEGs) with a q-value (p-adjusted via FDR/Benjamini–Hochberg correction) less than 0.1. The DEseq2 design included a grouping variable to test the interaction between sex and genotype, and the log2 fold changes were calculated with lfcShrink using the default apeglm package [35]. We did not perform batch-effect correction because RNA isolation, library preparation, and sequencing were performed in a single batch. MA-plots were generated by ggpubr (v0.4.0), and Eulerr diagrams were generated by eulerr (v6.1.1). Boxplots and scatterplots were generated by ggplot2 (v3.3.2). Gene ontology (GO) analyses were performed by Metascape [36] at http://metascape.org (accessed on 31 March 2022). We performed GO analysis on upregulated and downregulated genes separately, as this approach improves the identification of relevant ontologies [37]. The codes used in this study are available at https://github.com/umich-iwase-lab/2022_Kdm5cMalevsFemale (accessed on 31 March 2022). Raw sequencing data is deposited at Gene Expression Omnibus, GSE206346 https://www.ncbi.nlm.nih.gov/geo/query/acc.cgi?acc=GSE206346 (accessed on 31 March 2022).

### 2.3. Behavioral Paradigms

Eighty-one adult female mice (21 wild type (WT), 13 *Kdm5c*-HET, 29 *Kmt2a*-HET, and 16 double mutant (DM)), at least 4 months old at the beginning of experiments, underwent behavioral testing. Prior to behavioral testing, mice were acclimated to the animal colony room for at least one week, singly housed in standard cages, and provided with a standard lab diet and water ad libitum. A 12-h light–dark cycle (7:00 a.m.–7:00 p.m.) was maintained at 20 ± 2 °C. All testing was conducted by experimenters masked to genotype. Mice were tested in batches as they became available, with no differences in performance across batches. All mice were tested in all behavioral tasks. To avoid confounds between tests, stressful tasks (context fear conditioning and resident intruder tests) were performed last.

### 2.4. Contextual Fear Conditioning

Foreground context fear conditioning was assessed as previously described [13,38]. Mice were placed into a distinct context with white walls (9 ¾ × 12 ¾ × 9 ¾ in) and a 36-steel-rod grid floor (1/8 in diameter; ¼ spaced apart) (Med-Associates, St. Albans, VT, USA) and allowed to explore for 3 min, followed by a 2-s 0.8 mA shock, after which mice were immediately returned to their home cages in the colony room. Twenty-four hours later, mice were returned to the context, and freezing behavior was assessed with an NIR camera (VID-CAM-MONO-2A) and VideoFreeze (MedAssociates, St Albans, VT, USA). Data were analyzed using a one-way, between-groups ANOVA with genotype as the between-subjects factor, and Bonferroni corrections for post-hoc tests were used to correct for multiple comparisons. 

### 2.5. Three-Chambered Social Interaction

Mice were placed into a three-chambered apparatus consisting of one central chamber (24 × 20 × 30 cm^3^) and two identical side chambers (24.5 × 20 × 30 cm^3^), each with a mesh enclosure (8 cm diameter; 18 cm height; grey stainless-steel grid 3 mm diameter spaced 7.4 mm apart). All mice were habituated to the apparatus without other mice for 10 min, 24 h prior to the test. In the interaction test, a 2- to 3-month-old stranger female mouse (C57BL/6N) was placed in the mesh enclosure on one side of the box (“stranger”), and a toy mouse approximately the same size and color as the stranger mouse on the other (“toy”). Exploration of either the stranger or toy was defined as nose-point (sniffing) within 2 cm of the enclosure and used as a measure of social interaction [13]. Animals that did not interact with the stranger mouse at all were excluded from the analyses (*n* = 2). Behavior was automatically scored by Ethovision XT9 software as described above, and social preference was defined as time exploring strangers/total exploration time. Social preference was analyzed using one-sample *t*-tests for each genotype, and groups were compared using a one-way ANOVA, with Bonferroni-corrected post-hoc tests being used to correct for multiple comparisons across genotypes. 

### 2.6. Social Dominance Tube Test

Mice were habituated to a clear plastic cylindrical tube (1.5 in diameter; 50 cm length) and allowed to explore and enter the tube for 10 min, 24 h prior to testing. The test was conducted by placing two mice of different genotypes at opposite ends of the tube and allowing them to walk to the middle. The trial was terminated when one mouse (submissive mouse) retreated until it exited the tube with all four paws. This was scored as a loss for the submissive mouse and a win for the mouse that remained in the tube (the dominant mouse). Each mouse was tested against three different opponents, each of a different genotype, counterbalanced across groups. Videos were recorded by Ethovision XT9 software as described above, and videos were manually scored by trained experimenters with genotypes masked. The number of “wins” was reported as a percentage of the total number of matches. Data were analyzed using an Exact Binomial Test with 0.5 as the probability of success (win or loss). 

### 2.7. Resident-Intruder Test

To test whether *Kdm5c-HET*, *Kmt2a-HET*, or DM females exhibited aggression by fighting, a behavior that is unusual in virgin *mus musculus* females, we conducted the resident intruder test as previously described [13]. We observed no fighting by any mouse of any genotype (data not shown).

## 3. Results

### 3.1. Kdm5c-Heterozygous Female Brains Show Apparent Gene Expression Changes Similar to Male Mutants

To assess the role of lysine demethylase 5C (KDM5C) in neurodevelopment across sexes, we performed bulk mRNA sequencing (mRNA-seq) of postnatal day 6 (P6) forebrains, encompassing the cortices and hippocampi. The genotypes of animals were wild-type (WT, both male and female), *Kdm5c* hemizygous knockout male (*Kdm5c*-KO), and *Kdm5c* heterozygous female (*Kdm5c*-HET) mice. We have previously demonstrated that targeted deletion of mouse exons 11 and 12 abolishes KDM5C’s enzymatic function and protein production [14]. We first confirmed the absence of mRNA-seq reads of exons 11 and 12 in *Kdm5c*-KO males and approximately half the expression in *Kdm5c*-HET females compared to their sex-matched WT controls (Figure 1A). We next assessed *Kdm5c* expression in WT males and females, as *KDM5C* is known to escape X-inactivation in multiple species and at varying degrees across tissues [39,40,41]. Indeed, we found the *Kdm5c* expression in female forebrains to be approximately 1.5-fold higher than in male brains (Welch’s *t*-test, *p* = 1.15 × 10^−5^) (Figure 1B). Principal component analysis (PCA) demonstrated that sex rather than *Kdm5c* genotypes accounts for transcriptome variability overall (Figure 1C). 

We next evaluated the impact of *Kdm5c* loss on the male and female transcriptome by identifying differentially expressed genes (DEGs) with DESeq2 (q < 0.1). *Kdm5c*-KO males showed 708 genes upregulated and 371 genes downregulated compared to WT males. Notably, gene expression changes were also evident in *Kdm5c*-HET females, albeit with lower DEG numbers, 122 up and 14 down, compared to males (Figure 1D,E). Most DEGs are upregulated in the mutants, consistent with KDM5C being a transcriptional repressor [14]. In our previous RNA-seq study of the adult (4–8 months) male *Kdm5c*-KO hippocampus with identical experimental procedures, mutation, and analytical pipeline, 271 genes were upregulated, and 73 genes were downregulated [13]. Thus, the total DEG number of the male *Kdm5c*-KO P6 forebrain was >3-fold larger than the adult hippocampi, suggesting a pronounced vulnerability of neuronal maturation processes to *Kdm5c* loss. While most upregulated genes were shared between *Kdm5c*-HET females and *Kdm5c*-KO males, some genes showed significant changes only in one sex (Figure 1F). To assess the degree of dysregulation between sexes, we then plotted the log2 fold change (Log2FC) of shared and uniquely upregulated DEGs in *Kdm5c*-KO males versus *Kdm5c*-HET females. We found that the DEGs with the greatest Log2FC from WT were those dysregulated in both sexes (Figure 1G, left), and females exhibited 40% of the extent of male dysregulation of these shared genes (Figure 1G, right). 

Some of the common DEGs between sexes are implicated in known phenotypes of human *KDM5C* patients. For example, in both sexes, the DEG with the greatest Log2FC is *Gap junction protein, beta 1* (*Gjb1*) (Male: Log2FC = 1.96, q = 1.92 × 10^−82^; Female: Log2FC = 1.17, q = 4.01 × 10^−37^) (Figure 1H). *Gjb1* encodes connexin-32, whose expression increases across postnatal cortical development [42] to form gap junctions in oligodendrocytes and neurons [43] and is implicated in epilepsy [44]. Another shared DEG of interest is *cyclin-dependent kinase inhibitor 1c* (*Cdkn1c*) (Male: Log2FC = 0.89, q = 8.02 × 10^−20^; Female: Log2FC = 0.472, q = 1.48 × 10^−5^) (Figure 1I). *Cdkn1c* is a paternally imprinted gene involved in neocortical development [45] and aggressive behavior [46]. 

These results indicate the following. First, KDM5C heterozygosity is sufficient to cause readily detectable gene expression changes in females. Second, genes relevant to known manifestations of KDM5C deficiency are commonly misregulated between male and female mutants. Third, the degree of dysregulation of the common genes was milder in *Kdm5c*-HET females than in *Kdm5c*-KO males. The last point provides insight into sex chromosome evolution (See Section 4). 

### 3.2. Kdm5c Deficiency Also Leads to Sex-Specific Gene Expression Changes

In addition to the common gene expression changes between the sexes, we also found 847 male-specific DEGs and 20 female-specific DEGs (q < 0.1, Figure 1F,G). To gain insights into the sexually dimorphic consequences of KDM5C deficiency, we performed gene ontology (GO) analysis on DEGs upregulated in *Kdm5c*-KO males or *Kdm5c*-HET females (708 and 122 genes, respectively) with Metascape [36]. Many ontologies related to cell projection and adhesion were significantly enriched in *Kdm5c*-KO males (Figure 2A). Contrastingly, up DEGs for *Kdm5c*-HET females are enriched for ontologies relating to cellular development (Figure 2B, such as “regulation of neural precursor cell proliferation” (GO:2000177, *p* = 0.0004). While there were not enough downregulated DEGs in *Kdm5c*-HET females (n = 14) to perform ontology analysis, *Kdm5c*-KO male downregulated DEGs had significant ontologies that positively regulate synapse formation and function (Figure 2A)—consistent with our previous findings that male *Kdm5c*-KOs have decreased dendritic spine density in the hippocampus and amygdala [13].

Many male-specific DEGs derepressed in *Kdm5c* mutants have known roles in aggression and memory. For example, *phenylethanolamine N-methyltransferase* (*Pnmt*) methylates norepinephrine to form epinephrine [47], and *Prmt* overexpression causes extreme aggression in male mice [48] (Figure 2C). Additionally, the “neuroactive ligand-receptor interaction” KEGG pathway (mmu04080, *p* = 0.0004) is enriched only in *Kdm5c*-KO males (Figure 2A). *Kdm5c*-KO DEGs in this gene ontology have diverse roles in influencing memory, including the neuropeptide *galanin* (*Gal*) (Figure 2C). Galanin is involved in widespread neuromodulatory roles in sleep, nociception, metabolism, mood, and cognition, by inhibiting excitatory neurotransmission [49,50].

In females, key development-related genes contributed to the female-specific ontology enrichment. One gene of interest is *midkine (Mdk)*, with a slight but significant upregulation in *Kdm5c*-HET females but not in *Kdm5c*-KO males (Figure 2D, Table 1). *Mdk* is a neurotrophic factor expressed transiently during embryogenesis to promote neurite outgrowth [51,52]. Another female-specific DEG, *JPX transcript*, *XIST activator* (*Jpx*) is a long-noncoding RNA (lncRNA) crucial for X-inactivation, a females-specific epigenetic process [53,54] (Figure 2D, Table 1). *Jpx* is upregulated as both male and female embryonic stem cells differentiate, but the gene promotes X-inactivation only in females by inducing *Xist* expression [53,54]. All differentially expressed up- and downregulated genes found in males and females are listed in Appendix A.

Altogether, our transcriptomic data demonstrate that loss of *Kdm5c* in males and females has a sexually dimorphic impact on a subset of genes. Upregulation of early developmental genes represents the female mutant, while aggression and memory-related genes represent the male mutant.

### 3.3. Kdm5c and Double Kdm5c and Kmt2a Mutations Result in Social Behavior Deficits in Females

In the 3-chambered social interaction task (Figure 3D,E), both WT and *Kmt2a*-HET females showed a significant preference for interaction with the stranger mouse over the toy mouse (one-sample *t*-test WT: *p* = 0.046, *Kmt2a*: *p* = 0.012), DMs showed no preference (*p* = 0.068), and *Kdm5c*-HETs showed a significant avoidance of the mouse and preference for the toy (*p* < 0.001). This pattern was also evident in comparing between genotypes, where a significant overall effect of genotype (F(3,78) = 8.053, *p* < 0.001, η_p_^2^ = 0.244) was driven by a significant decrease in social preference by *Kdm5c*-HET females compared with both WT (*p* = 0.001) and *Kmt2a* (*p* < 0.001). DMs, despite showing no social preference and a trend towards avoidance, were not different from any other genotype, albeit with a trend towards decreased preference compared to *Kmt2a* mice (DM vs. WT: *p* = 0.125; vs. *Kmt2a*: *p* = 0.057; vs. *Kdm5c p* = 0.497) Figure 3D). Importantly, this is not driven by total interaction time. Across genotypes, animals interacted with stranger or toy a similar amount of time (time stranger + time toy; F(3,78) < 1) (Figure 3E). This pattern suggests that both *Kdm5c*-HET and DM females are impaired in social preference but that *Kmt2a* knockdown in *Kdm5c*-HETs nevertheless may decrease avoidance of the stranger mouse. 

A different pattern emerged in the social dominance tube test; DM but not *Kdm5c*-HET female mice showed a submissive phenotype compared with WT mice (*p* < 0.001), with no differences between WT, *Kmt2a*-HET, *Kdm5c*-HET female mice (all *p* = 1) (Figure 3F). This result was in contrast to patterns observed in males where, as previously described, both *Kmt2a*-HET and *Kdm5c*-KO males showed increased dominance over WT, whereas DM males, like the female DMs, showed a submissive phenotype [13]. 

Together, these results demonstrate that *Kdm5c*-HET females show deficits in specific aspects of social behaviors, most notably avoidance of other animals without a change in social dominance. Although *Kmt2a*-HET mice showed no overt changes in either social preference or dominance behaviors, the female double *Kdm5c*-HET, *Kmt2a*-HET mutants showed modifications, reducing avoidance and—similar to the male DM mice—driving a striking increase in submissive behaviors.

### 3.4. Double Kmt2a and Kdm5c Mutations Mediate Context Fear Memory Impairment in Females

We next used behavioral tasks to examine the impact of *Kdm5c* deficiency and its restoration by *Kmt2a* heterozygous deletion on context fear memory and social behaviors in female mice. To this end, we compared WT, *Kdm5c*-HET, *Kmt2a*-HET, and double mutants (*Kdm5c*-HET, *Kmt2a*-HET, DM) on foreground context fear conditioning (Figure 3A–C). We observed a significant main effect of genotype on Context Fear Conditioning (F(3,81) =5.08, *p* = 0.003, η_p_^2^ = 0.165), in which only DM females showed significantly lower freezing level compared with WT (*p* = 0.014) and *Kmt2a*-HET females (*p* = 0.003) (Figure 3C). In females, neither the partial loss of *Kdm5c* nor *Kmt2a* showed impaired memory compared with WTs (both *p* = 1.0). There were no significant differences across genotypes in freezing or locomotor activity during training (both F(3,81) < 1) (Figure 3A). Nevertheless, we did observe a significant effect of genotype in response to shock (F(3,81) = 7.46, *p* < 0.001. η_p_^2^ = 0.225), driven by significantly higher locomotor activity burst in response to footshock in *Kmt5c*-HET females (*cf* WT: *p* = 0.02; *cf Kmt2a*: *p* < 0.001). DM females did not show this effect (cf WT: *p* = 0.31; cf *Kmt2a*: *p* = 0.12) (Figure 3B). 

This pattern of deficits in female DM, but not in Kdm5c-HET females, are somewhat surprising. We and others have observed impaired context fear conditioning in Kdm5c-KO males, and a rescue effect in double Kdm5c-KO, Kmt2a-HET males [13]. In females, we observed no significant deficits in Kdm5c-HETs, and substantially worse performance in DM mice in contextual fear conditioning. Importantly, the increased shock response in Kdm5c-HETs may indicate increased sensitivity of Kdm5c-HET females to shock that may compensate for a subtle memory impairment effect. Nevertheless, these results suggest a somewhat different interaction of histone methylation dynamics in memory processes in female animals compared to our previous male study.

## 4. Discussion

In this work, we characterize the transcriptome of female *Kdm5c-HET* mice in the developing brain and determine the effects of *Kdm5c* loss and *Kmt2a-Kdm5c* antagonisms on behavior in females. Our data indicate that the loss of *KDM5C* leads to both common and distinct gene expression changes between sexes, and these changes may underlie differential patterns across sex of behavioral traits in *KDM5C* disorder mouse models. 

Commonly dysregulated genes in *Kdm5c*-deficient males and females may explain the behavioral traits seen in mice and human patients. Many MRXCSCJ patients have epilepsy [26], and *Kdm5c*-KO male mice have an increased propensity for kainic acid-induced seizures [31]. Our study highlights *Gjb1,* which encodes the protein connexin-32 that forms gap-junctions in oligodendrocytes and neurons [43], as a potential mediator of seizures (Figure 1H). Most studies on *Gjb1* focus on loss-of-function mutations, as they cause the sensorimotor neuropathy Charcot–Marie–Tooth type 1 [67]. Meanwhile, *Gjb1* mRNA and connexin-32 protein levels increase when epileptic activity is induced by bicuculline in the hippocampus [44]. Such a *Gjb1* overexpression may exacerbate seizure propensity with KDM5C deficiency. Two other characteristics of MRXSCJ and its mouse model are short stature and hyper aggression [22]. *Cdkn1c*, overexpressed in *Kdm5c*-deficient male and female developing forebrains, is a good candidate for potentially contributing to these two traits (Figure 1I) for the following reasons. Reminiscent of patients and mice with *KDM5C* mutations, microduplications of the region harboring *Cdkn1c* cause Silver-Russell Syndrome, characterized by growth retardation and short stature [68,69]. Mice overexpressing *Cdkn1c* are small [70], and the male transgenic mice display hyper aggression in the social dominance tube test [46]. Currently, future studies are warranted to test the contribution of these candidate genes. 

Genes whose expression changed uniquely in male or female *Kdm5c*-mutant mice illuminate the sexually-dimorphic manifestations of KDM5C deficiency. As female-specific DEGs, we highlighted *Jpx* and *Mdk*, both involved in early embryogenesis (Figure 2). In particular, *Jpx* is a key player in X chromosome inactivation, a dosage compensation mechanism unique to female cells. KDM5C also regulates the expression of *Xist*, another critical lncRNA in X-inactivation in early embryogenesis [30]. The female-specific dysregulation of developmental genes such as *Jpx* and *Xist* may explain why *Kdm5c*-homozygous female embryos are lethal [30]. Meanwhile, some male-specific DEGs in *Kdm5c* mutants, such as *Prmt* and *Glanin*, have known roles in MRXSCJ-impaired behaviors, such as aggression and memory (Figure 2C). *Pnmt* encodes an epinephrine-synthesis enzyme, and transgenic overexpression of this gene in mice results in elevated fighting behaviors in males but no changes in female behavior [48]. Galanin has an important role in learning and memory—galanin overexpression in the mouse forebrain impairs spatial learning [71,72,73], and this gene is overexpressed in the limbic system of post-mortem Alzheimer’s disease patients [74]. These results imply that common traits between sexes, such as aggression and memory, may involve distinct mechanisms downstream of *Kdm5c* loss. However, our gene expression study was performed in P6 animals, and behavior was assessed in adults; thus, developmental contribution of gene misregulation awaits future validations. Regardless, the female-specific gene expression changes call attention to possible unique symptoms in MRXSCJ female patients. 

This pattern of common and distinct dysregulation was also observed at the level of behavior. Whereas *Kdm5c*-HET females showed intact foreground fear conditioning, the double *Kmt2a-Kdm5c* knockout impaired memory performance. This pattern contrasts with the impaired memory in *Kdm5c*-KO and rescue in the double mutants we previously observed in male animals [13]. Female *Kdm5c*-HET mice—with and without *Kmt2a* mutation—also exhibited dysregulation of social behaviors. *Kdm5c-HET* mice showed social avoidance of a novel mouse instead of preference, and the *Kmt2a-Kdm5c* double mutants also failed to show social preference. Unlike in males, in which either *Kmt2a* or *Kdm5c* loss triggered hyperaggression [13], in females, neither *Kmt2a*-HET nor *Kdm5c*-HET alone altered social dominance. Nevertheless, consistent with that observed in males, the double mutant females showed a strikingly submissive phenotype. 

The *Kdm5c*-HET females exhibited a milder hippocampal memory deficit compared with males, consistent with the weaker gene misregulation in female mice (Figure 1). Intriguingly, other studies have demonstrated that females do show impairments in the background—or cued—context fear conditioning tasks [31]. The foreground context fear conditioning protocol used here is less sensitive to disruption by hippocampal manipulations [75,76,77] than those that include a discrete cue, and females, in particular, may be more likely to recruit non-hippocampal circuits for context fear conditioning [38,77]. It is not that females are impervious to disruption, however. Unlike male *Kdm5c-Kmt2a* double mutants, which showed no fear memory deficits, female DMs were more impaired than their *Kdm5c*-HET counterparts. 

Together, these findings demonstrate an overlapping but distinct pattern of gene misregulation and behavioral deficits in male *Kdm5c*-KO and female *Kdm5c*-HET mice that mimics the sex differences in symptomatology in individuals with MRXSCJ. Indeed, as with patients, some symptoms—including memory impairments and aggression—are less severe in female mice. These data suggest that other behaviors—and in particular social anxiety—may be more evident in girls with MRXSCJ. Importantly, these data also suggest that treatments targeting *KDM5C* or compensatory methylation enzymes may need to take sex into account carefully. In these studies, we have observed that whereas *Kmt2a* knockdown exerts a partial rescue for some behavioral deficits in males, the same genetic manipulation appears to unmask deficits in females. Although it remains likely that developmental effects play some role in the adverse effects of genetic intervention here, understanding sex differences in gene expression and behavioral symptomatology in MRXSCJ individuals will be essential for optimizing therapeutic interventions and strategies for both girls and boys.

Our work also provides insights into sex chromosome evolution. *Kdm5c* escapes X inactivation [39,40,41] (Figure 1). A unique feature of X-inactivation escapees is the presence of their paralogues in the Y chromosome [78,79]. KDM5D is the Y-linked paralogue for KDM5C in mice, humans, and other mammals [20]. Sex chromosomes are thought to originate from ordinary autosomes, and Y-chromosome genes have continued to be lost during the deterioration of Y. The persistence of the Y-linked paralogues of X-inactivation escapees led to the hypothesis that the X–Y pairs play the same roles that require a high dose achieved by expressing both X- and Y-genes, like many other autosomal genes [78]. This hypothesis predicts that *Kdm5c* loss in males will have the same impact as *Kdm5c* heterozygosity in females because KDM5C + KDM5D = 2x KDM5C. However, our data are discordant with this prediction. The unisexual DEGs showed more pronounced dysregulation in *Kdm5c*-KO males than in *Kdm5c*-HET females; KDM5D could not compensate for the loss of KDM5C (Figure 1G). These data render the above hypothesis less tenable for this X–Y pair. Instead, our data suggest that KDM5C and KDM5D have distinct functions in behavior and brain gene expression. 

## Figures and Tables

**Figure 1 cells-12-00637-f001:**
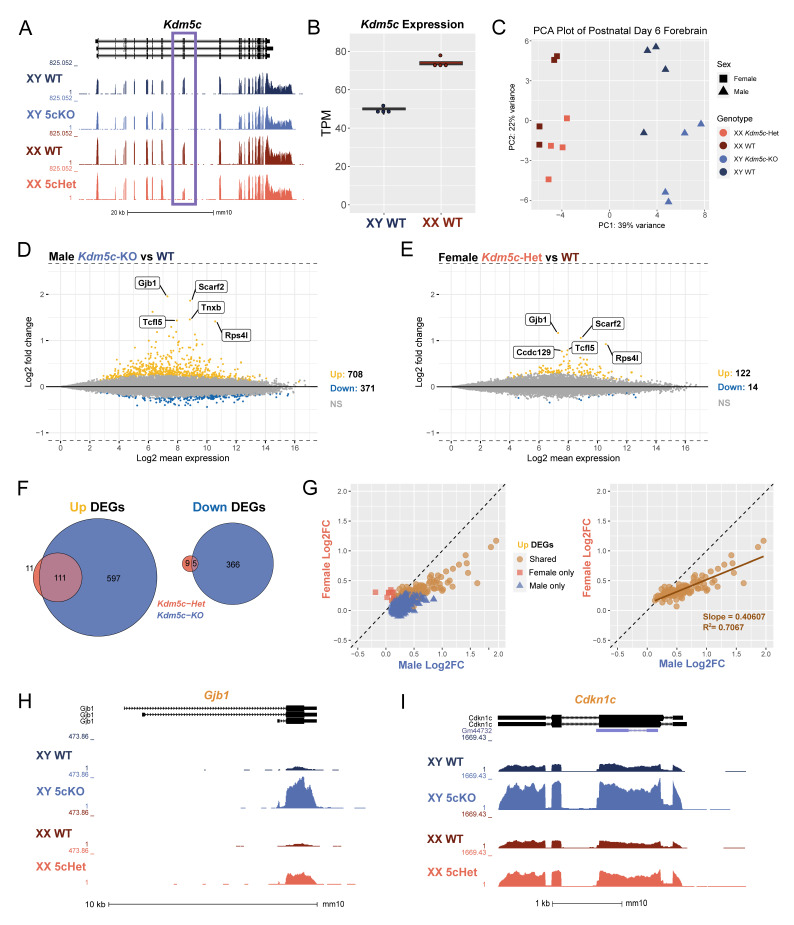
Loss of *Kdm5c* results in dose-dependent transcriptional dysregulation in the male and female mouse brain. (**A**). UCSC genome browser view displaying the transcription of *Kdm5c* in the P6 forebrain of wild type male (XY WT), *Kdm5c* knockout male (XY 5cKO), wild type female (XX WT), and *Kdm5c* heterozygous (XX 5cHet). Purple box highlights deletion of exons 11 and 12 in 5cKO and half reduction in 5cHet. (**B**). Comparison of *Kdm5c* RNA expression between wild-type male and female samples as transcripts per million (TPM). (**C**). Principal component analysis (PCA) of all mRNA-seq replicates. (**D**,**E**). MA plots of DESeq2 results from male and female brains with *Kdm5c* loss. Differentially expressed genes (DEGs), defined as a q-value < 0.1. DEGs upregulated from WT in yellow (Up) and downregulated from WT in blue (Down). Gray is non-significant genes (NS). Top 5 Up DEGs based on log2 fold change are annotated. (**F**). Overlap of male and female upregulated and downregulated DEGs, with male *Kdm5c*-KO DEGs in blue and female *Kdm5c*-Het DEGs in pink. (**G**). Left—Scatter plot of Up DEGs based on their log2 fold change from wild type in males (x-axis) or females (y-axis). Blue triangles are DEGs upregulated only in *Kdm5c*-KO males, pink squares are DEGs upregulated only in *Kdm5c*-HET females, and orange circles are shared DEGs upregulated in both males and females. Right—Same as left but displaying only shared DEGs. (**H**,**I**). UCSC genome browser view highlighting shared DEGs for males and females, *Gjb1* and *Cdkn1c*.

**Figure 2 cells-12-00637-f002:**
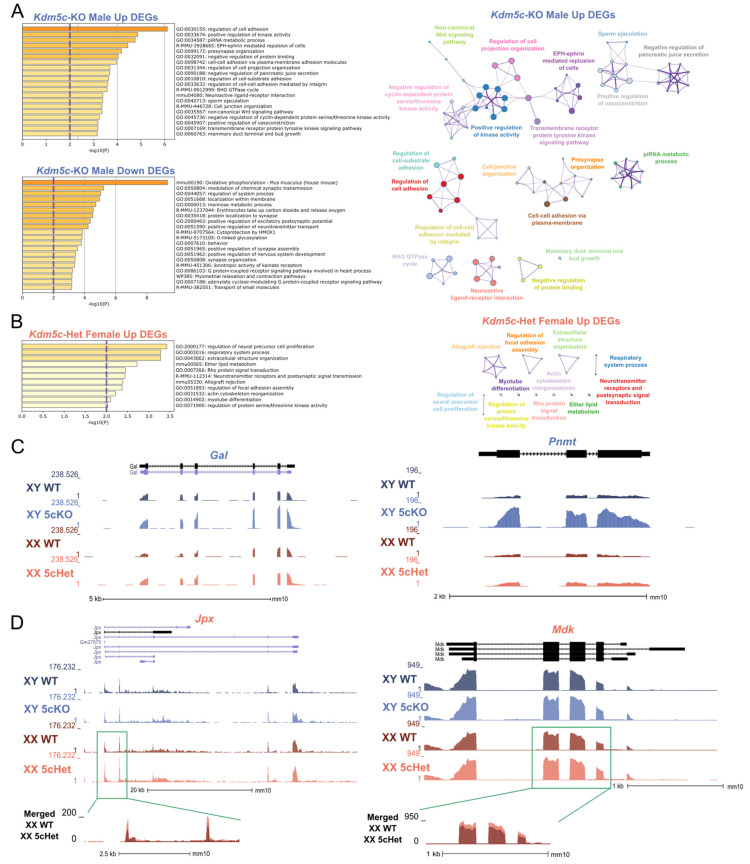
*Sexually dimorphic changes in genes related to development, aggression, and memory with loss of Kdm5c*. (**A**,**B**). Top gene ontology (GO) terms from Metascape for DEGs in *Kdm5c*-KO males (**A**) and *Kdm5c*-HET females (**B**). The purple dashed lines indicate the significance threshold. Right—Metascape GO enrichment map. Colors: enrichment networks, nodes: gene ontologies, edges: ontologies with a Kappa similarity greater than 0.3, thicker edges representing a greater Kappa similarity. (**C**). UCSC genome browser view of two genes, *Gal* and *Pnmt*, uniquely upregulated in *Kdm5c*-KO males (XY 5cKO) but not in *Kdm5c*-HET females (XX 5cHet). (**D**) Two genes, *Jpx* and *Mdk*, are uniquely upregulated in XX 5cHET but not in XY 5cKO.

**Figure 3 cells-12-00637-f003:**
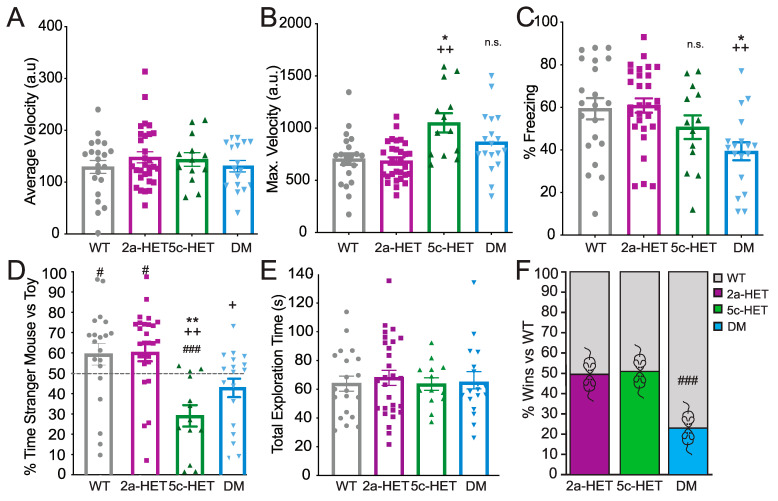
Altered memory and social phenotypes in Kdm5c-Het female mice are not rescued—and may be exacerbated—in double mutant mice. (**A**–**C**) Foreground context fear conditioning. (**A**) There were no differences across genotypes in locomotor activity during training. (**B**) *Kdm5c*-HET females showed exaggerated locomotor response to footshock during context fear conditioning. (**C**) DM but not *Kdm5c*-HET females showed deficits in a foreground context fear conditioning paradigm. (**D**) In contrast to context fear conditioning, both *Kdm5c*-HET and DM females showed deficits in social preference in the three-chambered task. *Kdm5c*-HET mice showed a robust aversion to the stranger and preference for a toy mouse, whereas DM mice exhibited no preference for, and also no aversion to, the stranger mouse. (**E**) Differences across genotypes were not due to differences in exploration—all groups showed equivalent total exploration of stranger and toy mouse combined. (**F**) DM females, as previously observed in males, showed striking submissive behavior in the Social Dominance Tube Test. Neither *Kdm5c*- nor *Kmt2a*-HET females were different from wildtypes. # *p* < 0.05, ### *p* < 0.001 vs. 50%; * *p* < 0.05, ** *p* < 0.01 vs. WT, + *p* = 0.057, ++ *p* < 0.01 vs. *2a*-HET, n.s. = non-significant. N = 21 (WT), 13 (*Kdm5c*-HET), 29 (*Kmt2a*-HET), and 16 (DM): all 4–6-month-old adult females.

**Table 1 cells-12-00637-t001:** Eleven genes significantly upregulated in the *Kdm5c-HET* female but not in the *Kdm5c*-KO male forebrain.

MGI Symbol	Full Name	Ensembl	Base Mean	XX log2FC	XX padj	XY log2FC	XY padj	Feature Type	Known Functions
*Arhgef19*	Rho guanine nucleotide exchange factor (GEF) 19	ENSMUSG00000028919	98.585	0.346	0.021	0.088	0.739	protein coding gene	Not much known in the brain, but SNPs are implicated in language impairments [55].
*Gm46620*	predicted gene, 46620	ENSMUSG00000118012	165.056	0.305	0.061	−0.185	0.253	lncRNA gene	Gene module with no known function
*Mdk*	midkine	ENSMUSG00000027239	331.224	0.305	0.028	0.116	0.546	protein coding gene	Growth factor highly expressed during early brain development [56]. Promotes neurite outgrowth and survival [57].
*Cfap100*	cilia and flagella associated protein 100	ENSMUSG00000048794	122.031	0.305	0.098	0.140	0.493	protein coding gene	Unknown function, but structurally similar to proteins involved in cilia and flagella motility [58].
*H2-Ab1*	histocompatibility 2, class II antigen A, beta 1	ENSMUSG00000073421	15.660	0.302	0.011	0.050	0.804	protein coding gene	Influences antigen processing and presentation via MHC class II [59]. No known brain function.
*Jpx*	Jpx transcript, Xist activator (non-protein coding)	ENSMUSG00000097571	154.681	0.297	0.088	0.048	0.866	lncRNA gene	Long non-coding RNA that induces Xist expression for X-inactivation [53,54].
*Dppa5a*	developmental pluripotency associated 5A	ENSMUSG00000060461	18.336	0.234	0.041	0.139	0.392	protein coding gene	Highly expressed in embryonic stem cells and germ cells and thought to play a role in pluripotency [60,61].
*Rhov*	ras homolog family member V	ENSMUSG00000034226	498.992	0.220	0.098	0.021	0.936	protein coding gene	Atypical, constitutively active GTPase implicated in neural crest development [62].
*Trim71*	tripartite motif-containing 71	ENSMUSG00000079259	18.205	0.196	0.068	0.151	0.166	protein coding gene	Plays important roles in embryonic neurogenesis [63] and postnatal ependymal cells [64].
*Rcn1*	reticulocalbin 1	ENSMUSG00000005973	1336.228	0.189	0.083	0.143	0.127	protein coding gene	Calcium binding protein located in the endoplasmic reticulum [65].
*Car11*	carbonic anhydrase 11	ENSMUSG00000003273	2481.408	0.169	0.039	0.103	0.227	protein coding gene	Homologous to Car10, potential extracellular binding partner of neurexin that influence synaptogenesis [66]

## Data Availability

The RNA-seq data are available in NCBI’s Gene Expression Omnibus (Accession number: GSE206346).

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
