# Peer review of "Sexually Dimorphic Alterations in the Transcriptome and Behavior with Loss of Histone Demethylase KDM5C"

_cells, 2023, doi:10.3390/cells12040637_

Round 1

Reviewer 1 Report

The manuscript by Bonefas et al. analyzes a mouse model of Claes-Jensen syndrome and directly compares gene expression profiles and behaviors of hemizygous and heterozygous mutant animals. This manuscript addresses important question regarding Kdm5C and its effects in females. The experiments presented are clear and rigorous, and their conclusions are supported by the data shown. I have only a couple of comments about the manuscript. 

OMIM has changed the official name of MRXCCJ to avoid the use of the problematic “mental retardation” phrase. It is now “Intellectual developmental disorder, X-linked syndromic, Claes-Jensen type.

What are the protein levels observed in KDM5C heterozygous mice compared to wild type? 

Is it possible to examine X chromosome inactivation bias in the KDM5C heterozygous mice (in the brain)? This has typically been hard to study in human patients leading to blood cells primarily being used to examine X-chromosome inactivation and KDM5C’s escape of this.

In interpreting the phenotypes in which KDM5C heterozygous mice are combined with Kmt2a heterozygotes, it would be useful to know if there are changes to the level of H3K4me2/3 globally or at specific genes in these genotypes? 

Sometimes the nomenclature used is confusing. For example, line 301, “Kdm5c-HET males” refers to Kdm5c hemizygous, Kmt2a heterozygous mice (I think). Whereas in the prior sentence “Kdm5c-HET” refers to heterozygous females.  

In Figure 2F, I find it difficult to see the increased expression across the Jpx gene in the Kdm5C heterozygotes. Perhaps it is less evident near the promoter (which I assume is the right hand side of the schematic) and more in the downstream exons? Similarly with the Mdk gene. This is likely due to the modest change in gene expression (the changes in hemizygous males are striking), but I wonder if there’s a better way to show that data so that it’s clearer? Is it possible to merge the tracks so that they are easier to directly compare? 

Minor:

On line 20, the authors mention ATRX. It would be useful to also include the fact that this gene, like KDM5C, is X-linked.

Author Response

Point 1: OMIM has changed the official name of MRXCCJ to avoid the use of the problematic “mental retardation” phrase. It is now “Intellectual developmental disorder, X-linked syndromic, Claes-Jensen type”.

Response 1: Thank you for mentioning this nomenclature change, we have updated the text accordingly.

Point 2: What are the protein levels observed in KDM5C heterozygous mice compared to wild type? 

Response 2: We have previously tried to ascertain KDM5C protein level differences in wild type males and females and agree this is an important facet of X-inactivation escape. However, we have encountered challenges in obtaining conclusive, quantitative results in detecting endogenous KDM5C protein likely due to KDM5C’s large molecular weight (>250kDa).

Point 3: Is it possible to examine X chromosome inactivation bias in the KDM5C heterozygous mice (in the brain)? This has typically been hard to study in human patients leading to blood cells primarily being used to examine X-chromosome inactivation and KDM5C’s escape of this.

Response 3: Using a mouse model of early embryogenesis, we found that initially, an even percentage of cells inactivate chromosomes harboring the mutant or wild-type Kdm5c However, over early development this bias shifts towards preferentially inactivating the mutant Kdm5c allele (Gayen et al. bioRxiv 2017, Extended Data Fig. 3B https://doi.org/10.1101/175174). Since this occurs very early in embryogenesis, we expect the brain to display similar patterns of X-inactivation skewing but have not rigorously tested this. An interesting future direction would be to cross mice with divergent SNP loci to see if there are brain regions more likely to have the mutant or wild-type allele active or if brain region-specific patterns correlate to individual mouse phenotypic severity.

Point 4: In interpreting the phenotypes in which KDM5C heterozygous mice are combined with Kmt2a heterozygotes, it would be useful to know if there are changes to the level of H3K4me2/3 globally or at specific genes in these genotypes?

Response 4: In male mice, Western blot of whole-brain lysates showed that global H3K4me1-3 are unchanged. However, via ChIP-seq we found differentially methylated H3K4 regions (DMRs) that are dysregulated in the opposite direction in Kdm5c or Kmt2a single mutants display rescue towards wild-type levels of methylation in double mutant mice (Vallianatos et al, 2020 Commun Biol https://doi.org/10.1038/s42003-020-1001-6). Because Kdm5c loss results in a more subtle phenotype in females for both behavior and RNA-seq, we have yet to do this experiment in female mice. However, we think this is a very important future direction that will help us determine the effectiveness of targeting Kmt2a for therapeutics in both sexes, particularly regarding the sexually dimorphic gene expression changes.

Point 5: Sometimes the nomenclature used is confusing. For example, line 301, “Kdm5c-HET males” refers to Kdm5c hemizygous, Kmt2a heterozygous mice (I think). Whereas in the prior sentence “Kdm5c-HET” refers to heterozygous females. 

Response 5: We fixed this error as well as clarified which sex we are referring to throughout the manuscript.

Point 6: In Figure 2F, I find it difficult to see the increased expression across the Jpx gene in the Kdm5C heterozygotes. Perhaps it is less evident near the promoter (which I assume is the right hand side of the schematic) and more in the downstream exons? Similarly with the Mdk gene. This is likely due to the modest change in gene expression (the changes in hemizygous males are striking), but I wonder if there’s a better way to show that data so that it’s clearer? Is it possible to merge the tracks so that they are easier to directly compare?

Response 6: Thank you for this suggestion, we have updated Figure 2 to include zoomed-in overlays of the XX WT and XX Kdm5c-HET bigwig tracks so that these subtle changes in gene expression are more readily visible.

Point 7: On line 20, the authors mention ATRX. It would be useful to also include the fact that this gene, like KDM5C, is X-linked.

Response 7: Thank you for pointing out this similarity, we have added this to the text.

Reviewer 2 Report

Thank you for the opportunity to review this manuscript entitled "Sexually dimorphic alterations in the transcriptome and behavior with loss of histone demethylase KDM5C". This is very interesting research to clarify novel associations between neurodevelopmental disorders and sex chromosomes.

Some abbreviations are missing the full term in the manuscript such as ATRX. The authors should check and revise the manuscript properly.

Author Response

Point 1: Some abbreviations are missing the full term in the manuscript such as ATRX. The authors should check and revise the manuscript properly. T

Point 2: Thank you for bringing these errors to our attention, we have edited the manuscript to include the full name of all abbreviations.

Reviewer 3 Report

Using a mouse model, the current study looked into the effects of KDM5C loss on the transcriptome and behavior. The authors also demonstrated that the loss of KDM5C is likely to have distinct effects in males and females, indicating that females need a larger dose of KDM5C. Overall, the present findings are interesting and may provide insights into the different prevalence of relevant disorders in man and women. A few aspects of statistical analysis may be improved, though. Details are provided below.

1. I’d suggest the authors to supply sufficient details for mRNA-seq section in METHOD. More specific questions regarding the Method are provided below:

1)    The current draft lack of some part of the pre-processing steps, including pairing, quality control (e.g., FastQC), and trimming steps.

2)    Did the authors apply the following steps before identifying those differentially expressed genes: 1) gene expression normalization for composition bias; 2) batch effect correction (if applies); 3) removal of genes with 0 read counts.

3)    Did the authors include any covariate when identifying those differentially expressed genes?

4)    If I understand correct, the “q-value” in Line #97 represents the adjusted p value after multiple testing correction. In that case, please clarify what type of multiple testing correction was applied.

5)    If any parameter was applied in any step, please specify it in METHOD.

2. The authors stated “In a previous RNA-seq study of adult KDM5C hippocampus (4-8 months) with identical experimental procedures and analytical pipeline, 271 genes were upregulated, and 73 genes were down-regulated [13]. Thus, the total DEG number of Kdm5c-KO P6 forebrain was > 3-fold larger than the adult hippocampi, suggesting a pronounced vulnerability of circuit maturation processes to Kdm5c loss.” Since the number of differentially expressed genes are different between males and females, it’d be great if the authors could specify the sample information for the cited study. This may enable the readers to grasp the comparison.

3. In Line #223-231, the authors performed the functional enrichment analysis on upregulated genes. It’s unclear to me why the authors focused on only upregulated genes, as I assume that the dysregulated genes may work together. For example, an upregulated gene A may increase producing a protein X, while a downregulated gene B may also improve the production of the protein X.

4. In the animal behavioral experiments, the authors applied pairwise tests among four groups without multiple testing correction.

5. In Line # 319-321, the authors described that “DMs, despite showing no social preference and a trend towards avoidance were not different from any other genotype, albeit with a trend towards decreased preference compared with Kmt2a mice (DM vs WT: p = 0.125; vs Kmt2a: p = 0.057; vs Kdm5c p = 0.497) …” However, the p values indicate that the DM group is not statistically different from ANY of the other three groups.

6. Please correct the typo in Line #321.

Author Response

Point 1: I’d suggest the authors to supply sufficient details for mRNA-seq section in METHOD. More specific questions regarding the Method are provided below:

1)    The current draft lack of some part of the pre-processing steps, including pairing, quality control (e.g., FastQC), and trimming steps.

2)    Did the authors apply the following steps before identifying those differentially expressed genes: 1) gene expression normalization for composition bias; 2) batch effect correction (if applies); 3) removal of genes with 0 read counts.

3)    Did the authors include any covariate when identifying those differentially expressed genes?

4)    If I understand correct, the “q-value” in Line #97 represents the adjusted p value after multiple testing correction. In that case, please clarify what type of multiple testing correction was applied.

5)    If any parameter was applied in any step, please specify it in METHOD.

Response 1: We have updated the mRNA-seq methods section to be more thorough in all steps of the data analysis. Briefly, the quality of the paired-end reads was assessed via FastQC and did not require trimming. Batch effect correction did not apply. We used the lfcShrink feature of DESeq2 to normalize read counts, including unexpressed and lowly expressed genes, using the default apeglm package settings. When setting up our DESeq2 results tables, we used a grouping variable to assess the interaction between sex and genotype as variables. We clarified that q-values refer to the adjusted p-values (padj) from DESeq2 that are calculated via the default FDR/Benjamini Hochberg correction. Additional information on DESeq2 analysis can be found here: DOI: 10.1186/s13059-014-0550-8. Additionally, we have uploaded all scripts used to complete each stage of the RNAseq data analysis into our github repository, which is also linked in the paper’s method section https://github.com/umich-iwase-lab/2022_Kdm5cMalevsFemale.

Point 2: The authors stated “In a previous RNA-seq study of adult KDM5C hippocampus (4-8 months) with identical experimental procedures and analytical pipeline, 271 genes were upregulated, and 73 genes were down-regulated [13]. Thus, the total DEG number of Kdm5c-KO P6 forebrain was > 3-fold larger than the adult hippocampi, suggesting a pronounced vulnerability of circuit maturation processes to Kdm5c loss.” Since the number of differentially expressed genes are different between males and females, it’d be great if the authors could specify the sample information for the cited study. This may enable the readers to grasp the comparison.

Response 2: We updated the text to include the information that our previous study was done in adult Kdm5c-KO males with the same mutation.

Point 3: In Line #223-231, the authors performed the functional enrichment analysis on upregulated genes. It’s unclear to me why the authors focused on only upregulated genes, as I assume that the dysregulated genes may work together. For example, an upregulated gene A may increase producing a protein X, while a downregulated gene B may also improve the production of the protein X.

Response 3: The reviewer raises an excellent point on upregulated and downregulated genes potentially acting in concert. We, therefore, also performed Metascape gene ontology analysis on the genes downregulated in Kdm5c-KO males and added them to Figure 2 and the text. Kdm5c-HET females only have 14 downregulated DEGs and thus did not have gene ontology enrichment. Additionally, we included downregulated DEGs for each sex in our supplemental table 1.

Point 4: In the animal behavioral experiments, the authors applied pairwise tests among four groups without multiple testing correction.

Response 4: In all experiments, with the exception of the exact binomial tests for SDTT, Bonferroni corrections following ANOVA were used to correct for multiple post-hoc tests. We followed only significant main or interaction effects from ANOVA with Bonferroni corrections – one of the more conservative corrections for multiple comparisons in common usage. We have now clarified this in the methods.

Point 5: In Line # 319-321, the authors described that “DMs, despite showing no social preference and a trend towards avoidance were not different from any other genotype, albeit with a trend towards decreased preference compared with Kmt2a mice (DM vs WT: p = 0.125; vs Kmt2a: p = 0.057; vs Kdm5c p = 0.497) …” However, the p values indicate that the DM group is not statistically different from ANY of the other three groups.

Response 5: We acknowledge that this can be a somewhat contentious issue: since p=0.05 is an arbitrary cutoff, a common question is how we think about statistical results that are barely different from that threshold. We agree with the reviewer that it does not reach the 0.05 threshold and, as such is not significantly different from Kmt2a-HET mice. However, since p=0.057, we also believe that it is worth noting a statistical trend, especially given the significant overall effect of genotype on this measure (F(3,78) = 8.053, p < 0.001).

Point 6: Please correct the typo in Line #321.

Response 6: We proof-read this line for accuracy.

Round 2

Reviewer 3 Report

1. R.E. the authors’ response 1: “We used the lfcShrink feature of DESeq2 to normalize read counts, including unexpressed and lowly expressed genes, using the default apeglm package settings.” What the authors described sounds like they removed the unexpressed and lowly expressed genes (i.e., quality control on genes) instead of normalization. As far as I understand, a typical DEG study include the following steps: 1) quality control on subjects and on genes; 2) correction for potential biases, e.g., batch effect correction; 3) Normalization; 4) DEG analysis with covariates controlled. It’d be great if the authors could double-check their pipeline and texts in METHOD.

2. R.E. the previous point 3 and response 3. In my initial comment, I recommended that the functional enrichment analysis be run on all DEGs collectively because upregulated and downregulated genes might cooperate. I did, however, just read several studies that suggested functional enrichment analysis on upregulated genes and on downregulated genes separately may be more effective and discover more underlying mechanisms/pathways. I concur with the authors' current approach, but it would be nice if they could add a sentence to the METHOD describing why functional enrichment analyses were carried out separately and citing pertinent articles.

Author Response

We appreciate the reviewers for further careful assessment of our manuscript. Please find below our responses to the comments.

Point 1: 1. R.E. the authors’ response 1: “We used the lfcShrink feature of DESeq2 to normalize read counts, including unexpressed and lowly expressed genes, using the default apeglm package settings.” What the authors described sounds like they removed the unexpressed and lowly expressed genes (i.e., quality control on genes) instead of normalization. As far as I understand, a typical DEG study include the following steps: 1) quality control on subjects and on genes; 2) correction for potential biases, e.g., batch effect correction; 3) Normalization; 4) DEG analysis with covariates controlled. It’d be great if the authors could double-check their pipeline and texts in METHOD.

Response 1: We apologize for any confusion. We did not remove any genes from our analysis, including unexpressed genes. lfcShrink is a normalization feature for calculating log2fold change from read counts with DESeq2. What we were highlighting is lfcShrink takes into account genes that are lowly expressed or unexpressed so that small, insignificant fluctuations in expression are not artificially inflated and called as differentially expressed. The paper describing the lfcShrink apeglm method can be found here: https://academic.oup.com/bioinformatics/article/35/12/2084/5159452 and we have now cited this paper in the method section.

For each step that the reviewer mentions: 1) We performed quality control using FastQC, and validated genotypes by analyzing the appropriate bigwig tracks 2) we did not require batch-effect correction because RNA isolation, library preparations, and sequencing were all done in a single batch. 3) normalization was done via lfcShrink’s apeglm setting 4) DEGs were called using DESeq2 contrasts with grouping variables that accounted for the interaction between sex and genotype. We ensured these steps are included in the provided github link and described in the methods. We amended the method section to clarify these points.

Point 2: 2. R.E. the previous point 3 and response 3. In my initial comment, I recommended that the functional enrichment analysis be run on all DEGs collectively because upregulated and downregulated genes might cooperate. I did, however, just read several studies that suggested functional enrichment analysis on upregulated genes and on downregulated genes separately may be more effective and discover more underlying mechanisms/pathways. I concur with the authors' current approach, but it would be nice if they could add a sentence to the METHOD describing why functional enrichment analyses were carried out separately and citing pertinent articles.

Response 2: We have included the paper below in the methods section as to why this approach is more apt for identifying biologically relevant ontologies.

https://pubmed.ncbi.nlm.nih.gov/24352673/